# Correlation Analysis of Evapotranspiration, Emissivity Contrast and Water Deficit Indices: A Case Study in Four Eddy Covariance Sites in Italy with Different Environmental Habitats

Michele Torresani [1,2,*], Guido Masiello [3], Nadia Vendrame [4], Giacomo Gerosa [5], Marco Falocchi [6], Enrico Tomelleri [1], Carmine Serio [3], Duccio Rocchini [7,8] and Dino Zardi [2,4]

1   Faculty of Science and Technology, Free University of Bolzano/Bozen, Piazza Università/Universitätsplatz 1, 39100 Bolzano/Bozen, Italy
2   Department of Civil, Environmental and Mechanical Engineering, University of Trento, 38123 Trento, Italy
3   School of Engineering, University of Basilicata, Via Ateneo Lucano 10, 85100 Potenza, Italy
4   Center Agriculture Food Environment (C3A), University of Trento, 38010 San Michele All'Adige, Italy
5   Department of Mathematics and Physics, Università Cattolica del Sacro Cuore, 25121 Brescia, Italy
6   Cisma S.r.l., Via Ipazia 2 c/o NOI Techpark, 39100 Bolzano, Italy
7   BIOME Lab., Department of Biological, Geological and Environmental Sciences, Alma Mater Studiorum University of Bologna, Via Irnerio 42, 40126 Bologna, Italy
8   Department of Spatial Sciences, Faculty of Environmental Sciences, Czech University of Life Sciences Prague, Kamýcka 129, 16500 Praha, Czech Republic
*   Correspondence: michele.torresani@unibz.it

**Abstract:** Evapotranspiration (ET) represents one of the essential processes controlling the exchange of energy by terrestrial vegetation, providing a strong connection between energy and water fluxes. Different methodologies have been developed in order to measure it at different spatial scales, ranging from individual plants to an entire watershed. In the last few years, several methods and approaches based on remotely sensed data have been developed over different ecosystems for the estimation of ET. In the present work, we outline the correlation between ET measured at four eddy covariance (EC) sites in Italy (situated either in forest or in grassland ecosystems) and (1) the emissivity contrast index (ECI) based on emissivity data from thermal infrared spectral channels of the MODIS and ASTER satellite sensors (CAMEL data-set); (2) the water deficit index (WDI), defined as the difference between the surface and dew point temperature modeled by the ECMWF (European Centre for Medium-Range Weather Forecasts) data. The analysis covers a time-series of 1 to 7 years depending on the site. The results showed that both the ECI and WDI correlate to the ET calculated through EC. In the relationship WDI-ET, the coefficient of determination ranges, depending on the study area, between 0.5 and 0.9, whereas it ranges between 0.5 and 0.7 when ET was correlated to the ECI. The slope and the sign of the latter relationship is influenced by the vegetation habitat, the snow cover (particularly in winter months) and the environmental heterogeneity of the area (calculated in this study through the concept of the spectral variation hypothesis using Rao's Q heterogeneity index).

**Keywords:** emissivity; evapotranspiration; heterogeneity; Rao's Q index; spectral variation hypothesis; thermal infrared

## 1. Introduction

Evapotranspiration (ET) is an important component of the forest hydrological budget, and influences the flow of water to downstream users, including aquatic habitats and human populations. Furthermore, it represents a considerable water loss in the landscape [1,2]. As an example, ET has been reported to inject into the atmosphere approximately 70% of annual precipitation in a loblolly pine (*Pinus taeda*) plantation in south-eastern USA [3], more than 85% in a Canadian black spruce (*Picea mariana*) forest [4] and more than 85% in a ponderosa pine (*Pinus ponderosa*) forest in Arizona [5]. Consequently, the magnitude and

seasonality of forest ET are important regulators of water resources available to humans and ecosystems. ET represents a crucial process within a broad range of systems, including ecology, hydrology and meteorology. For this reason, different methodologies have been developed in order to measure it at different spatial scales, ranging from individual plants to entire watersheds [6]. Various techniques have been developed to measure ET [6], including sap flow analysis [7], by weighing lysimeters [8], plant chambers, stable isotope [7,9], soil water budgets [10], land surface models [10] and eddy covariance (EC) [7,11]. More recently, remote sensing data, offering large area coverage, frequent updates and consistent quality, have been used in different studies to collect a quantitative information of ET over different ecosystems world-wide [12,13].

ET cannot be measured directly from remote sensing data. Indirect approaches [14], such as the energy balance approach [15], the Priestley–Talor approach [16,17] and through the use of spectral indices [18], are commonly applied. In general, process-based models that couple remote sensing information and ET have been widely used in science in the last several years, at both local and global scale. The models reproduce physical and plant physiological mechanisms that regulate ET, such as stomata processes, radiation absorption and water interception [14]. Different remote sensing approaches use land surface characteristics such as the leaf area index (LAI) and the albedo to estimate ET via surface energy balance or within-scene scaling [19,20]. Remote sensing thermal infrared measurements have also been largely used for the retrieval of ET information [21,22]. As an example, Hamberg et al.[23] illustrated the potential of thermal information derived from the ECOSTRESS satellite sensor for inferring land surface temperature and ET in different forest sites in Southern Ontario, Canada. Carlson et al. [24], again using the HCMM satellite, introduced a method for inferring different variables, including the distribution of evaporative fluxes and surface heat, in the cities of Los Angeles and St. Louis (USA). We refer to the following articles for an exhaustive overview of the use of infrared thermal radiation for ET retrievals [25–28]. For more general information about ET estimation techniques based on remote sensing data, Zhang et al. [12] provided an exhaustive review.

The thermal infrared (TIR) spectral region is also susceptible to soil moisture, allowing for the retrieval of the atmosphere's thermodynamic state along with the hydrometeorological conditions near the surface. The thermodynamic state close to the surface and the surface itself can be related straightforwardly to surface ET. A recent study by Masiello et al. [29] made use of the remote-sensed emissivity contrast index (ECI) based on TIR emissivity data derived from infrared atmospheric sounding interferometer (IASI) measurements [30,31] and demonstrated that it correlates with the water deficit index, or WDI, defined as the difference between the surface and dew point temperature close to it [32]. In [29], both the ECI and WDI have been obtained with a technique that enables the simultaneous retrieval of spectral emissivity and the vertical distribution of temperature (T), water vapor (Q) and other trace gases [33]. The WDI can be computed using in situ measurements or using modeled information, such as that of the European Centre for Medium Range Weather Forecasts (ECMWF) .

The ECI, firstly introduced by French et al. [34], in Masiello et al. [35] has been computed as the difference between the CAMEL emissivity channels (derived from the CAMEL database CAM5K30EM v002 [36–39]) at 8.6, 10.8 and 12.1 μm. The index was developed with an NDVI synergy to better classify vegetation cover and to overcome the limitations of the vegetation index, particularly in the discrimination of bare soil and senescent vegetation. It showed promising results in the classification of changes in land use when, for example, a vegetation regeneration follows the deforestation or forest degradation events [35]. The CAMEL dataset, where the emissivity information is stored, is produced by the combination of two distinct databases to take advantage of each product's characteristics. The first is the ASTER Global Emissivity (ASTER GEDv4), developed at the Jet Propulsion Laboratory (JPL): it has a temporal resolution of 1 month, a spatial resolution of 5 km and a spectral range from 8 to 12.0 μm. The MODIS baseline-t emissivity (MODBF) represents the second database: it is provided by the University of Wisconsin-Madison

and it has a spectral emissivity range from 3.6–12.0 μm. The resulting dataset is available globally in mean monthly time-steps with a spatial resolution of 5 km, with several layers providing information of emissivity (13 bands ranging from 3.6–14.3 μm), NDVI, snow fraction and related quality flags. The CAMEL dataset has been produced to design a uniform, long-term and calibrated emissivity database in order to advance the analysis of different applications, such as atmospheric retrievals and radiative transfer simulations. Within such a coarse spatial resolution, the environmental heterogeneity could be very high. For this reason, there is need for a sub-pixel heterogeneity assessment. The concept behind the spectral variation hypothesis (SVH) [40] could be used to assess the environmental heterogeneity within each pixel. This concept hypothesizes that the spectral response of a remotely sensed image could be used as a proxy to assess habitat heterogeneity and species diversity. Areas with a high spectral heterogeneity (SH) in a remotely sensed image have a high environmental heterogeneity with a higher number of available ecological niches. This concept was established firstly by Palmer et al. [40] and later developed by other authors [41]. The SVH has been tested in different ecosystems using various remote sensing data through the use of different SH indices. In the last few years, Rao's Q index (developed by Rao [42] for ecological purposes) has been proposed as an original SH measure [43] and has gained popularity due to the positive results obtained in various studies [44,45]. As stated by Rocchini et al. [43], "*given an image of N pixels, the Rao's Q is related to the sum of all the pixel values pairwise distances, each of which is multiplied by the relative abundance of each pair of pixels in the analyzed image*". Hence, Rao's Q index, in comparison to other heterogeneity indices, has the advantages of considering both the values (through the distance/difference between the pixel) and the abundance of the pixels in a considered image [46].

The main aim of this paper is to analyze the relationship between ET, derived from ground-based eddy-covariance (EC) surface measurements at four different sites in Italy, and both the ECI (based on emissivity data from the CAMEL database) and the WDI (based on the difference in the surface and dew-point temperature modeled by ECMWF data). In the first relationship, the effects of the snow cover, the different vegetations and the environmental heterogeneity (calculated through the concept of the SVH using Rao's Q index) were analyzed. The paper is organized as follows. Section 2 deals with data and methods. Results are shown in Section 3 and discussed in Section 4. Conclusions are drawn in Section 5.

## 2. Materials and Method

### 2.1. Study Areas

Four EC sites were used to assess the relationship between both ECI and WDI with ET.

The Renon site [47–49] is located in the province of Bolzano/Bozen in the Alps, in the municipality of Renon/Ritten at an elevation of 1740 m asl. The EC tower is located in a *Picea abies*-dominated forest (around 85 %), but also including *Pinus cembra* L., (12%) and *Larix decidua* Mill., (3%). The forest canopy is irregular, with maximal height of around 30 m. The annual average temperature is around 4.6 °C, and the average annual precipitation is approximately 900 mm.

The Monte Bondone site [50] is located on a mountain plateau (called "Viote del Monte Bondone") near the city of Trento at 1550 m asl. The mean annual air temperature is 5.5 °C and the mean annual rainfall is 1190 mm. The site is managed as productivity-extensive meadow, typical of the alpine regions, characterized by the presence of *Festuca rubra* (basal cover of 25%), *Nardus stricta* (13%) and *Trifolium* sp. (14.5%).

The Lavarone EC tower [47] is situated near the town of Lavarone in the province of Trento at an elevation of around 1350 m asl. The tower is located in uneven-aged mixed forest dominated by *Abies alba* (around 70%), *Fagus sylvatica* (15%) and *Picea abies* (15%). The forest canopy reaches an elevation of approximately 35 m. The average annual precipitation is approximately 1290 mm and the mean annual temperature is around 7.8 °C.

Finally, the Bosco della Fontana site [51,52] is located near the city of Mantova (at an elevation of 19 m asl.) in the middle of the Po valley, within a forest nature reserve of

around 235 ha. The wood canopy is 26 m high and dominated by *Carpinus betulus* L. and *Quercus robur* L. (57%), with a minor presence of *Acer campestre* L., *Prunus avium* L., *Fraxinus ornus* L. and *Ulmus minor* Mill., with *Alnus glutinosa* L. along the little rivers. The average annual precipitation is approximately 930 mm and the mean annual temperature is around 13.9 °C. The EC tower data are measured from a 42 m tall tower Figure 1.

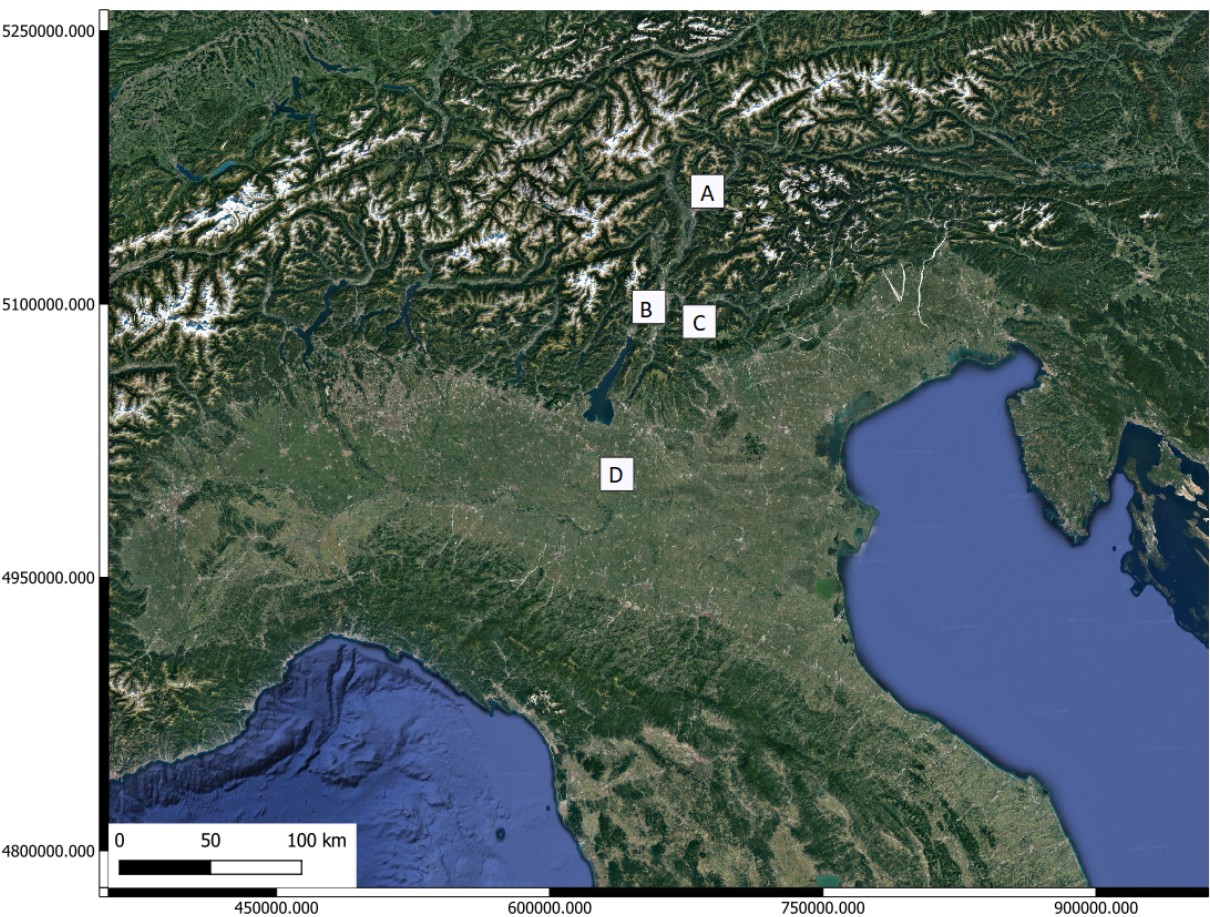

**Figure 1.** Google Earth image with the location of the four study sites in Italy. (A) Renon (Fluxnet code: IT-Ren); (B) Monte Bondone (Fluxnet code: IT-MBo); (C) Lavarone (Fluxnet code: IT-La2); (D) Bosco della Fontana (ICOS code: IT-BFt). CRS map: WGS84 /UTM zone 32N.

*2.2. Eddy Covariance Data*

Renon, Lavarone and Monte Bondone eddy covariance sites are part of the FLUXNET Network, where $CO_2$ fluxes, water vapour and other ancillary meteorological variables are measured at half-hourly intervals. Data are processed and quality controlled following the Fluxnet methodology [53]. Within the Fluxnet network, the data availability for the Renon site ranges from 1998 to 2013, for Lavarone, from 2003 to 2014 and, for Monte Bondone, from 2003 to 2013.

Bosco della Fontana is part of the ICOS Ecosystem Network. The eddy covariance tower measures half-hourly turbulent fluxes of $CO_2$, water vapor and different meteorological data following the ICOS protocols [54].

Despite the corrections applied in the calculation of EC fluxes, sensible and latent heat fluxes are usually underestimated at most EC sites with respect to the available energy at the surface [55], resulting in some uncertainty in the quantification of water lost by ecosystems through ET. For this reason, in our analysis, we used the latent heat flux adjusted by a correction factor based on the ratio between available energy and the sum of turbulent energy fluxes for each half hour [53]. The half-hourly latent heat flux data from the eddy

covariance sites obtained from the FLUXNET and ICOS datasets were converted to ET using "LE.to.ET" function of the "bigleaf" R package, applying the correlation parameter between depth units and energy of ET [56] (Formula (1)). The conversion was corrected using the half-hourly air temperature.

$$ET = LE/\lambda \tag{1}$$

where:
- ET is the evapotranspiration ($kg\ m^{-2}\ s^{-1}$);
- LE is the latent heat flux ($W\ m^{-2}$);
- $\lambda$ is the latent heat of vaporization $2.45\ MJ\ kg^{-1}$.

Successively, daily ET values were then accumulated and converted into monthly ET in order to be correlated to the ECI and WDI that were assessed on a monthly basis.

### 2.3. Emissivity Data and ECI Estimation

The ECI (that ranges in the interval [0, 1]) has been developed to discriminate between bare soil and vegetation [34] and to better classify vegetation cover [35]. For this present study, the methods introduced by Masiello et al. [35] were used to calculate the ECI from the CAMEL dataset. We used only the CAMEL pixels that had a "good" emissivity quality flag (value 1) in order to have an adequate overall accuracy.

The ECI is based on the channels at 8.6, 10.8 and 12.1 μm for the CAMEL dataset. According to different studies [34,35], these channels are indeed the most sensitive to bare, green and senescent vegetation. As a consequence, ECI is calculated as:

$$ECI = 1 - \delta\epsilon \tag{2}$$

where $\delta\epsilon$ represents the difference between the maximum and the minimum value of emissivity ($\epsilon$) among the three CAMEL spectral channels.

For each study area, monthly ECI was successively correlated to the monthly ET by a time series analysis and, successively, by linear regression. $R^2$ and p values were used to assess the strength and significance of the correlations. Due to the different temporal range data availability of ET and ECI, the correlations were tested differently for each study area. For the Renon study area, the time-series range from 2008 to 2013, for Lavarone, from 2008 to 2014, for Monte Bondone, from 2010 to 2013 and, for Bosco della Fontana, only the data from 2013 were available.

Furthermore, snow cover information, derived from the "snow fraction" layer of the CAMEL dataset, was included in the time-series correlation ET–ECI data. This layer provides information of snow cover on the basis of the normalized difference snow index (NDSI) [57], which ranges from 0 (no snow cover) to 100 (full snow cover). It is used to identify possible anomalies in the ET-ECI index correlation, particularly in the mountain sites (Renon, Lavarone and Monte Bondone), where snow remains on the ground for several winter months.

### 2.4. Meteorological Data and WDI Calculation

Monthly modeled data of surface temperature ($Ts_{ECMWF}$) and dew point temperature ($Td_{ECMWF}$) derived from the ECMWF [58] were used to compute the water deficit index, or WDI. ECMWF data were from the "Operational Analysis", and were released over a regular grid of $0.125° \times 0.125°$. For each EC site, the closest point of the ECMWF grid was chosen. Surface temperature ($Ts_{ECMWF}$) and dew point temperature ($Td_{ECMWF}$) were, respectively, the skin temperature and the 2 m dew point temperature from surface analyses.

WDI was then computed according to [32]:

$$WDI = Ts_{ECMWF} - Td_{ECMWF} \tag{3}$$

For this reason, WDI values depend on surface and dew point temperatures. High WDI values are expected in summer, especially in dry conditions, when the surface temperature becomes significantly higher than the dew temperature near it, whereas lower values are expected in winter.

Because of its definition and calculation, the WDI has a coarser spatial resolution than the ECI. However, the temperature and humidity fields are expected to be more homogeneous than the surface emissivity, which can have space scales of variability of a few meters or less.

As for the correlation ECI-ET, for each study area, the monthly WDI was successively correlated to the monthly ET by a time series analysis and, successively, by linear regression. $R^2$ and p values were used to assess the strength and significance of the correlations. Due to the different temporal range data availability of WDI and ET, the correlations were tested differently for each study area. In the Renon study area ,the correlation range was from 2010 to 2013, for Lavarone, from 2010 to 2014, for Monte Bondone, from 2010 to 2013 and, for Bosco della Fontana, only for 2013.

### 2.5. Assessment of the Environmental Heterogeneity

In order to assess the effect of the environmental heterogeneity within the ECI pixel (Figure 2), the SVH was assessed through Rao's Q index (Formula (4)) using an NDVI MODIS image (resolution of 500 m) captured on 8 June 2014. The choice of this date is related to the work of Torresani et al. [59], where they stated that the NDVI at this time of the year (summer), when it reaches the highest seasonal values, is more able to capture small variations in reflectance of different vegetation and, thus, of different ecosystems. For this purpose, the R-package function "Rao" of the R package *rasterdiv* [60] was implemented to retrieve a Rao's Q value for the single ECI pixel.

$$Q_{rs} = \sum_{i=1}^{F-1} \sum_{j=i+1}^{F} d_{ij} * p_i * p_j, \tag{4}$$

where:

- $Q_{rs}$ is the Rao's Q applied to remote sensing data;
- $p$ is the relative abundance of a pixel value in a selected study area (*F*). In our case, it is the CAMEL pixel;
- $d_{ij}$ is the distance between the *i*-th and *j*-th pixel value ($d_{ij} = d_{ji}$ and $d_{ii} = 0$);
- *i* and *j* identify two pixels within the area *F*.

The relative abundance $p$ was calculated as the ratio between the considered pixel ($p_i$ and $p_j$) and the total number of pixels in *F*. The distance matrix $d_{ij}$ can be built in different dimensions, allowing for the consideration of more than one band or raster at a time. In our case, the $d_{ij}$ was calculated as a simple Euclidean distance based on the NDVI image.

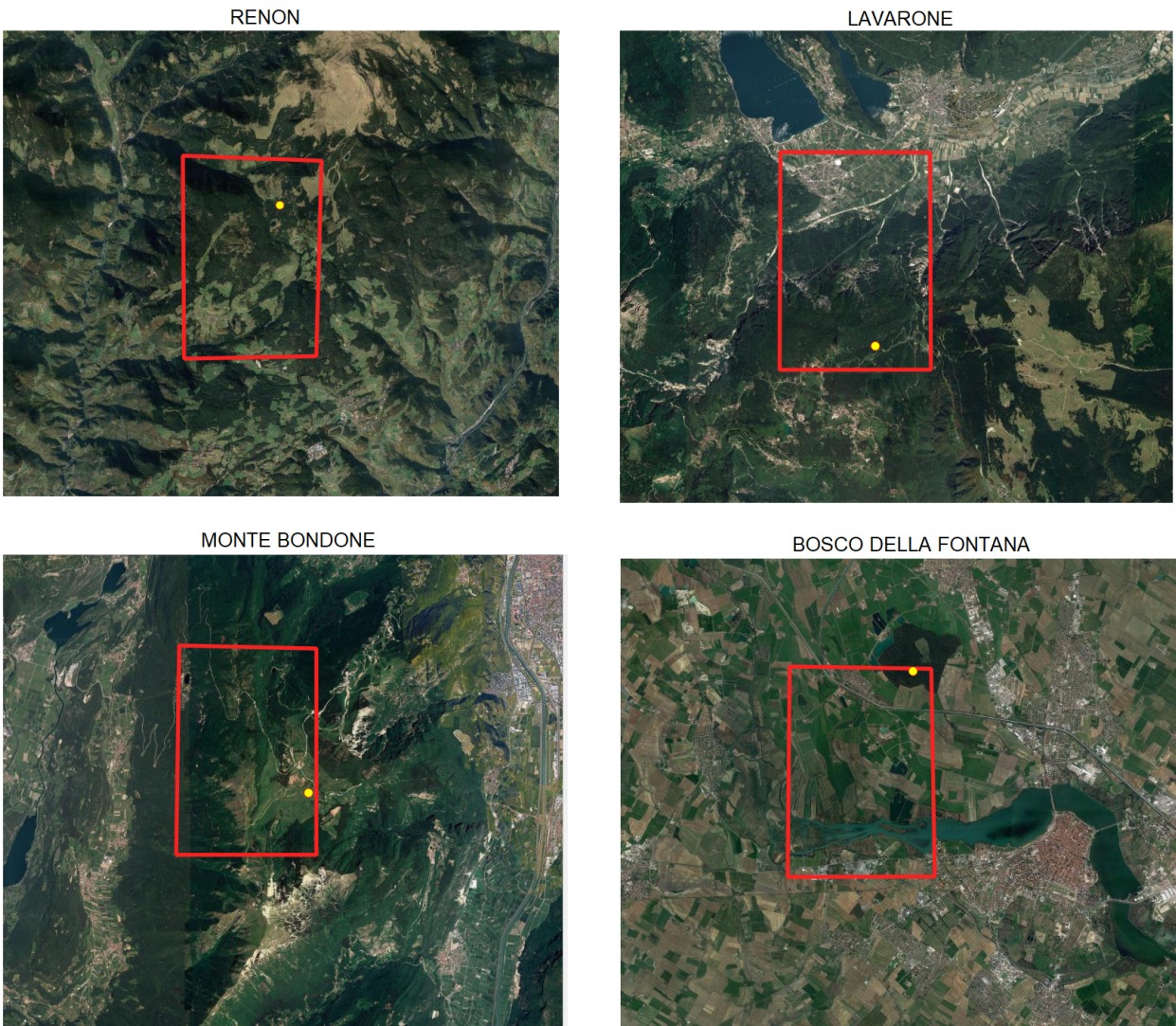

**Figure 2.** Google earth image with the location of the four eddy covariance towers (yellow points) for each site. In red, the CAMEL pixel used to derive the ECI. Date picture: Lavarone and Monte Bondone 6 December 2017; Renon 19 October 2017; Bosco delle Fontana 18 June 2013.

## 3. Results

### 3.1. Seasonal Evolution and Correlations

The temporal evolution of the ET and the monthly ECI with reference to the snow cover information is shown in Figure 3. For the Renon, Lavarone and Monte Bondone study area, both the ET and ECI seem to follow the same seasonal evolution, particularly in the summer months, when both the curves reaches the peak. In several situations, generally in the cold months, the ECI behaves differently, creating a "second peak" that does not follow the normal ET trend. The snow cover information explains the "winter peak" of the ECI, where the amplitude and the size of both the blue curve and black histogram are indeed similar. On the other hand, in the Bosco della Fontana study area, ET and the ECI have an opposite seasonal evolution. ET has a normal seasonal trend with the highest values in summer, decreasing in winter, whereas the ECI has its low values in summer.

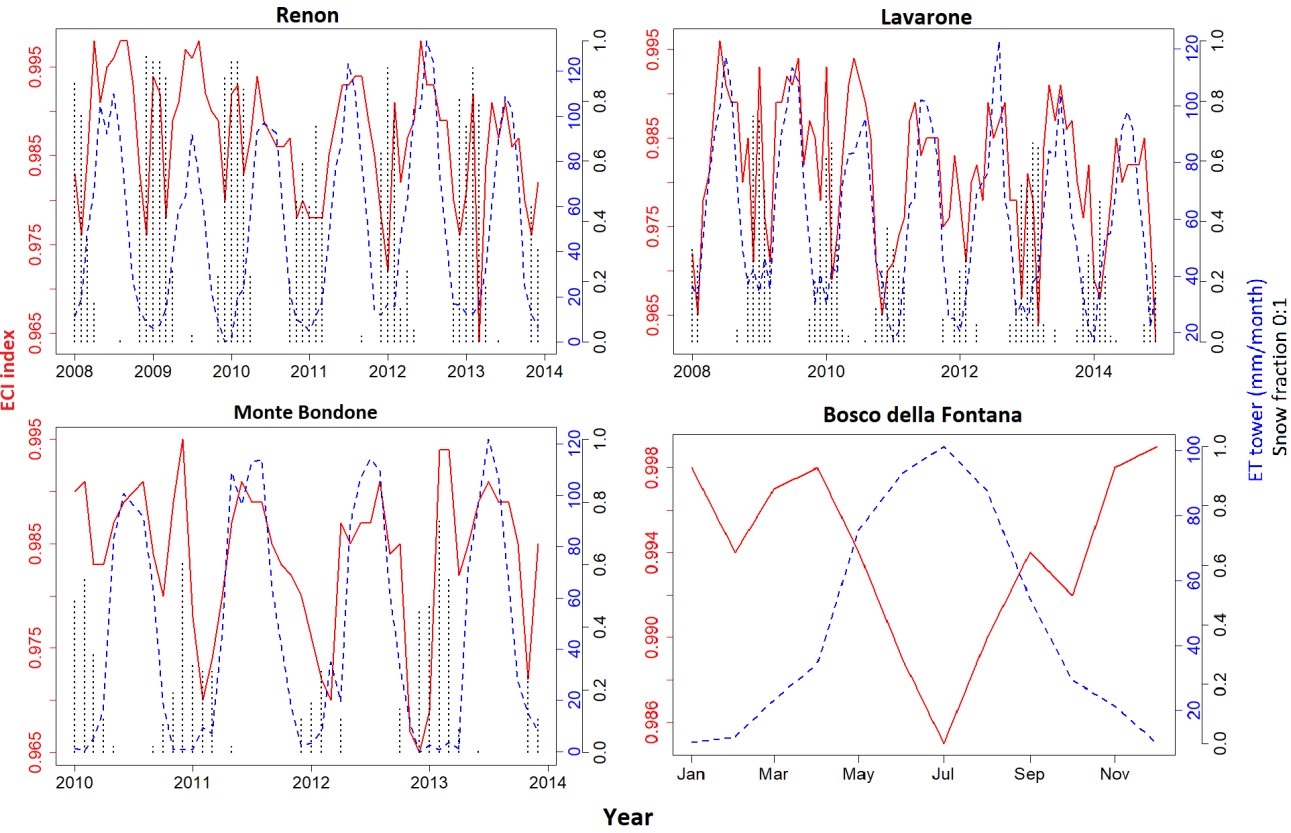

**Figure 3.** Relationship between the ET (ET tower—blue line) and the monthly ECI (red line).

Figure 4 shows the linear regression between ET-ECI considering the months with low snow cover (lower than 50%). This was carried out to assess the strength of the correlation without the "anomalous peaks" created by the ECI in the cold months. In the Renon, Lavarone and Monte Bondone study areas, the correlations are all positive, reaching an $R^2$ between 0.49 (Lavarone) to 0.68 (Renon). In the Bosco della Fontana study area, the correlation is negative, reaching an $R^2$ of 0.72.

The monthly time series of ET and the WDI are shown in Figure 5. In the four study areas, both the curves show the same trend, with the peak in the summer months and a lower value in winter.

Figure 6 shows the linear regression between ET and the WDI for the four considered study areas. $R^2$ ranges from 0.48 (Lavarone) to 0.89 (Bosco della Fontana). Unlike the correlation ET-ECI, the slope remains positive for all of the considered areas.

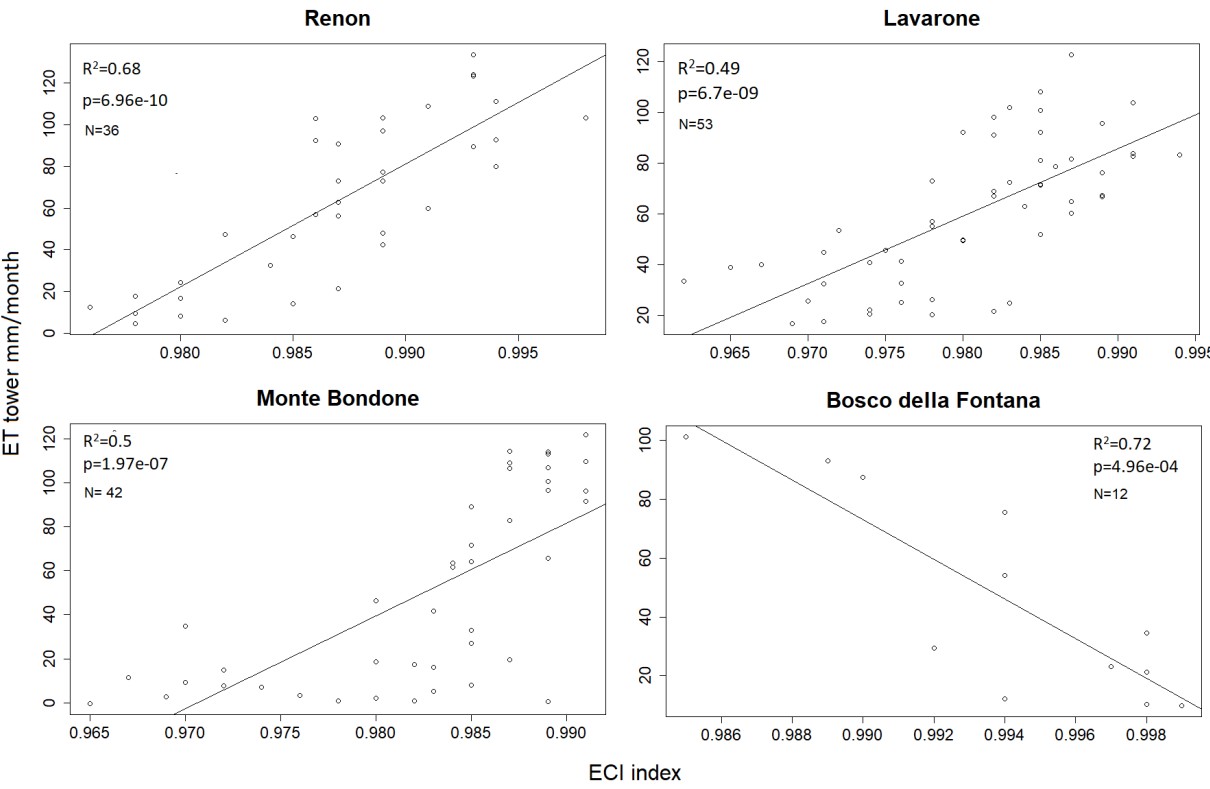

**Figure 4.** Linear regression between the ET (ET tower) and the monthly ECI when the snow cover is lower than 50%. *N* is the number of data points.

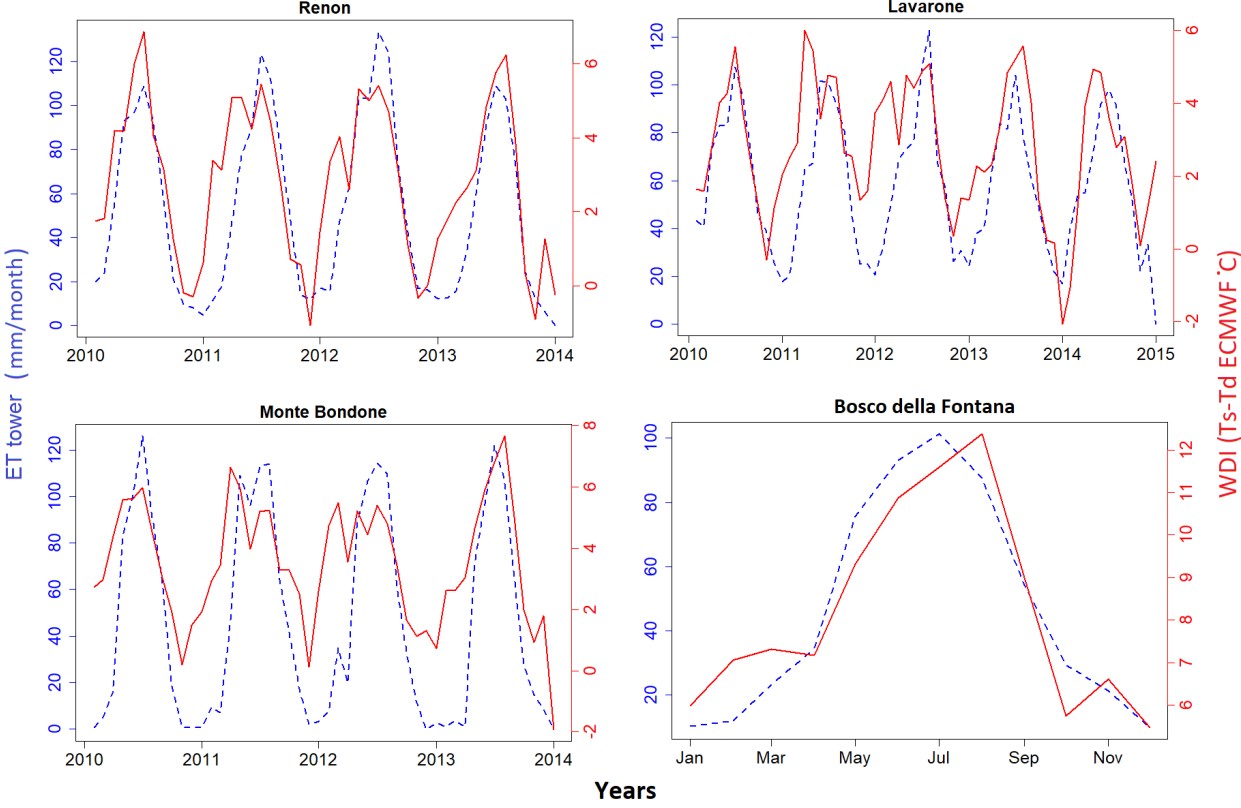

**Figure 5.** Time series of ET (blue dashed line) and WDI (red line)

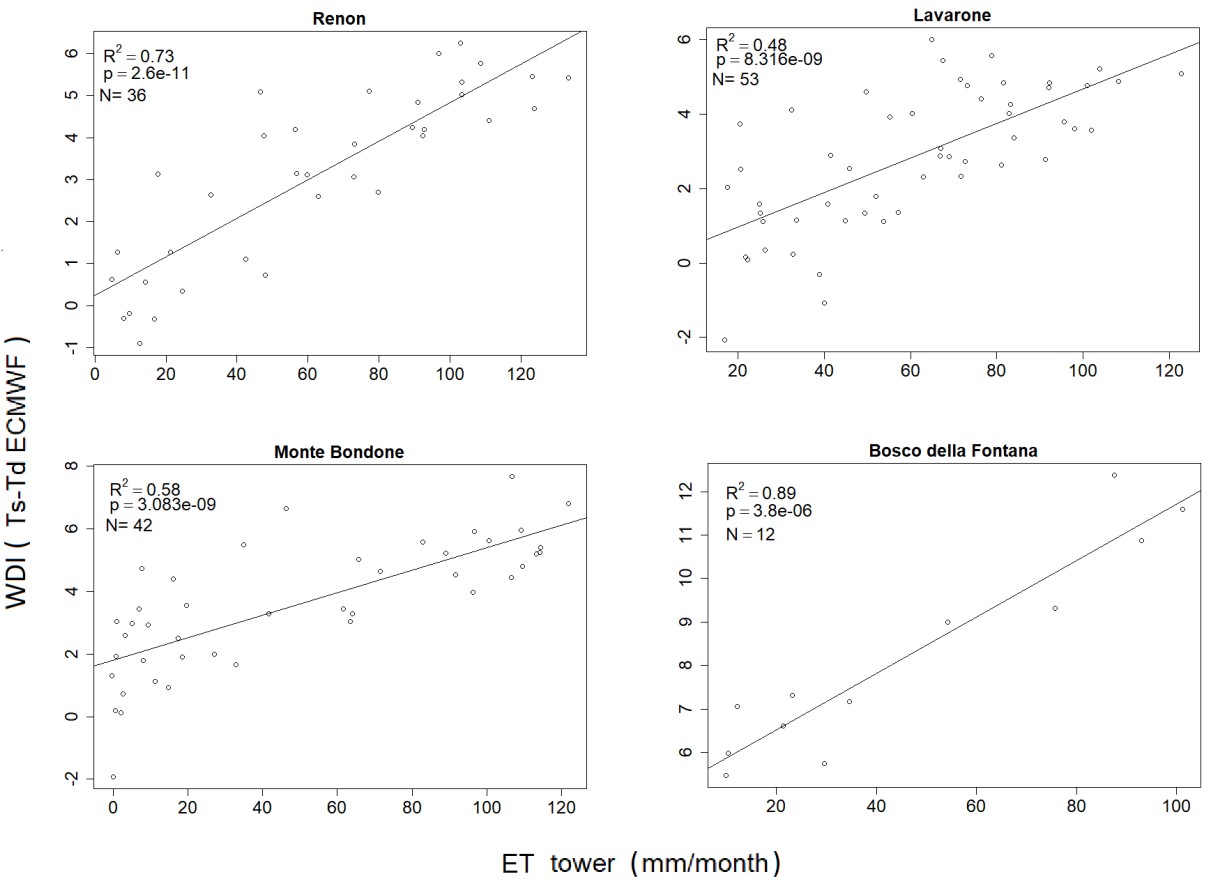

**Figure 6.** Linear regression between ET and WDI for the four considered study areas.

### 3.2. Environmental Heterogeneity

In order to understand the impact of the environmental heterogeneity, especially within the ECI pixel, the SH, as an indicator of habitat heterogeneity, was calculated through Rao's Q index over the four study areas. Table 1 summarizes Rao's Q index values for each site.

**Table 1.** Rao's Q value over the four ECI pixels of each study area.

| Area | Rao's Q Index |
|---|---|
| Monte Bondone | 0.036 |
| Renon | 0.061 |
| Lavarone | 0.075 |
| Bosco della Fontana | 0.082 |

The highest Rao's Q value is in the area of Bosco della Fonata (higher environmental heterogeneity), whereas the lowest is in the area of Monte Bondone (lower heterogeneity).

Figure 7 shows both the above mentioned study areas with the exact position of the eddy covariance tower (yellow dot) and the pixel size of the CAMEL database (red rectangle). Also from a first view, it is possible to notice the difference in habitat heterogeneity between the two sites. In the area of Monte Bondone (Figure 7A), the forested and grassland area are predominant. In the area of Bosco della Fontana (Figure 7B), the environmental heterogeneity is higher. Different habitats fall within the CAMEL pixel: the forested area (in the upper right corner), grassland, farmland, uncultivated areas and the aquatic ecosystems of the Mincio river and of the Mantova's lakes.

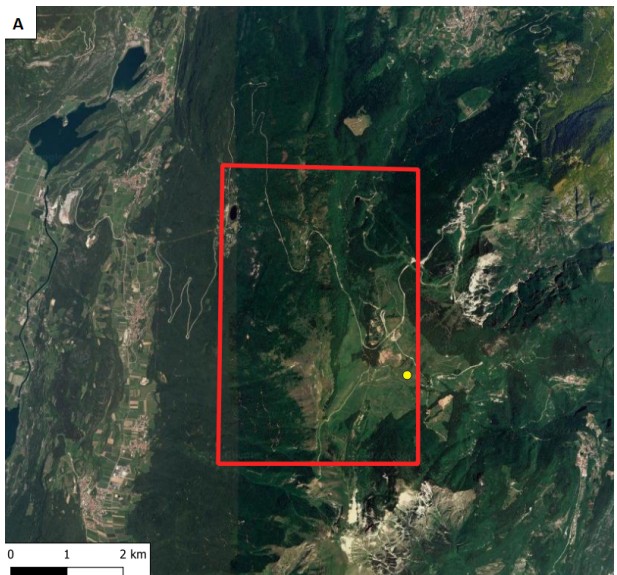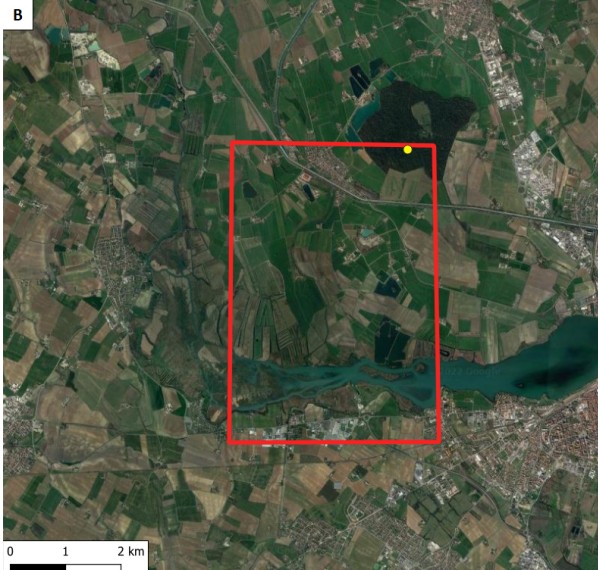

**Figure 7.** (**A**) shows the area of Monte Bondone and (**B**) shows the area of Bosco della Fontana; the yellow dot shows the position of the eddy covariance tower. The red rectangle shows the size of the CAMEL pixel.

## 4. Discussion

In this paper, we presented the correlation between the ECI (based on emissivity data from infrared spectral channels of the CAMEL dataset) and the WDI (based on the data from ECMWF analyses) against ET measured in four EC sites in Italy (Renon, Monte Bondone, Lavarone and Bosco della Fontana). Furthermore, the behavior of the above mentioned correlations in relation to the vegetation cover was discussed. Additionally, the environmental heterogeneity was assessed (using NDVI MODIS images and analyzed through the SH Rao's Q index) in order to evaluate the performance, particularly of the ECI in relation to the habitat fragmentation over the four study areas.

The results show that the correlation between ECI and ET is statistically significant for the four study areas. In the areas of Renon, Lavarone and Monte Bondone, the relationships are positive and they are influenced by the snow cover (estimated with the "snow fraction" layer of the CAMEL dataset), which interferes with the emissivity values. In the winter months, when the snow covers the earth surface, the ECI yields a second peak that does not follow the ET trend. As described by Masiello et al. [29,35], the ECI is highly influenced by the dryness and by the land cover. As an example, the ECI has a value of around 0.98 for water-covered surfaces (e.g., snow), whereas, for dry and senescent vegetation, the index reaches lower values. The ECI was initially developed to build synergy with the NDVI to overcome the drawback of the vegetation index that, in certain cases (as in the case of Masiello et al. [35] over the Congo basin area for a temporal range of seventeen years), is not able to discriminate senescent vegetation and bare soil; in particular, when the vegetation regeneration start after a deforestation or degradation event [35]. On the other hand, in the area of Bosco della Fonatana, the correlation ECI–ET was negative: in the considered year, the curves indeed had an opposite trend. Several reasons can explain this different trend: since the ECI is influenced by the dryness and by the land cover, its values could be distorted by the humidity of the soil and by the presence of surface water, which is very high in this area, located near the aquatic ecosystems of the Mincio river and of the Mantova's lakes. The other reason is related to the high habitat fragmentation within the CAMEL pixel that alters the ECI value (as shown in in Figure 7) and summarized in Table 1. The results of the table indeed show that the Rao's Q index, used to estimate the environmental heterogeneity (by the assessment of the SH), has the highest value in the area of Bosco della Fontana.

The concept of the SVH and the related environmental heterogeneity is therefore crucial not only in the estimation of biodiversity and the assessment of species diversity [41,61,62], but also in all studies where the ground information (e.g., energy fluxes) is correlated with the remote sensing data. This aspect is particularly common when using remote sensing information. An image, through the diversity of the various pixels, provides knowledge about the remote sensing response caused by the physical interaction of the measured information (emissivity in our case) with the Earth's surface. Thus, the raster grid of pixels that build up an image, represents just an average response of a real information. The question regarding the size of the pixels, in order to describe a certain area or to characterize a real situation or an event is, as previously stated, still debated in the research community [63,64]. Like in our case, images with a coarse spatial resolution tend to integrate the information of various subjects (e.g., vegetation, human artefacts, rivers.), homogenizing the signal and causing difficulties toward clearly identifying boundaries between spatial entities (individuals, vegetation types, ecosystem types) [65]. On the other hand, a fine spatial resolution may lead to a level of details within spatial entities that may cause a strong heterogeneity, leading to strong noises and uncertainties [66].

As far as the relationship of ET vs. WDI is concerned (Figures 5 and 6), we have found that the correlation is less sensitive to the vegetation changes and environmental heterogeneity. The slope of the linear regression remains positive, although we have explored sites with a very large heterogeneity. The difference in temperature ($Ts_{ECMWF}$—$Td_{ECMWF}$) is correlated with the ET because the surface temperature is strongly dependent on the impinging solar radiation, whereas the dew point temperature is dependent on both the air temperature and the humidity field. The partial mismatch between the two variables might be due to the lack of an additional meteorological parameter, such as the wind. The detail of the regression still seems to be site-dependent; however, in this case, there is no ambiguity that ET and the WDI are positive correlated. Since the WDI is based on temperatures, it is less influenced by the habitat fragmentation and by the environmental heterogeneity compared to the ECI, which is based on the emissivity of the surface. For this reason, the correlation holds true also in the area of Bosco della Fontana, which, as previously stated, showed an opposite result in the correlation ET-ECI.

Finally, we believe that the synergistic use of the ECI and WDI might increase the accuracy of ET estimation because of their different sensitivities to different aspects of the vegetation. The ECI is better suited to detecting changes in the vegetation state, green to senescent or transition to bare soil. These states can influence ET, but in a way that can be highly nonlinear. Conversely, the WDI is more linearly related to ET. The synergistic use of the two could be, e.g., of some interest during an intense heatwave, which has become common in temperate regions because of climate change (e.g., see https://climate.copernicus.eu/esotc/2021/globe-in-2021 (accessed on 8 May 2012)). In the event of heatwaves, we expect the WDI > 0 (e.g., see [32]), and a decreasing value of the ECI could show an early decay of the foliage to the senescent state, and hence vegetation stress.

## 5. Conclusions and Outlook

In this study, the correlation between data of ET derived from four eddy covariance sites in Italy (Renon in the Province of Bolzano, Monte Bondone and Lavarone in the Province of Trento and Bosco della Fontana in the Province of Mantova) and two indices— (1) the emissivity contrast index, or ECI; (2) the water stress index, or WDI—was assessed. Both indices were shown to correlate with in situ observations, which is good from the perspective of using remote-sensed data to monitor the state of vegetation from satellite. The correlation ECI-ET is influenced by the habitat heterogeneity and by the presence of snow/water in the surface. This could be critical, especially in areas covered by snow (e.g., mountain regions in winter), with surface water or with high environmental heterogeneity. The WDI showed generally fewer uncertainties in detecting the correct evolution of ET, in that the index is directly related to the thermodynamic parameters that govern ET and to the intensity of solar radiation. Furthermore, we believe that the synergistic use of

the ECI and WDI could lead to a more accurate ET estimation, bringing the benefits of both indices. Improvements on this side would also be greatly beneficial for providing a more accurate input to numerical weather and climate prediction models, for which, reliable estimates of fluxes over snow-covered terrain are still a challenging situation [67,68].

Further refinements can also be obtained from a more precise evaluation of ET from EC, taking into account timescales associated with different atmospheric conditions [69]. This is particularly applicable to the mountainous sites, where daily periodic flows, such as thermally driven slope wind and valley winds, are well established, and documented meteorological features of the mountain boundary layer [70–72]. This goal is among the scopes of the ongoing international cooperation effort TEAMx (Multi-Scale Transport and Exchange Processes in the Atmosphere over Mountains–Program and Experiment) through intensive field campaigns performed at selected target areas in the Alps, combining ground-based, airborne and remote sensing observations [73].

**Author Contributions:** Data curation, M.F.; Investigation, E.T. ; Methodology, G.G.; Supervision, D.R. and D.Z.; Writing—original draft, M.T.; Writing—review & editing, G.M., N.V. and C.S. All authors have read and agreed to the published version of the manuscript.

**Funding:** This research was carried out in the framework of the project 'OT4CLIMA', which was funded by the Italian Ministry of Education, University and Research (D.D. 2261 del 6.9.2018, PON R&I 2014-2020 and FSC). M.T. and D.R. were partially supported by the European Union's Horizon 2020 research and innovation program under grant agreement No 862480 (SHOWCASE). DR was partially supported by the Horizon Europe project EarthBridge and by the H2020 COST Action CA17134 'Optical synergies for spatiotemporal sensing of scalable ecophysiological traits (SENSECO).

**Data Availability Statement:** Not applicable.

**Conflicts of Interest:** The authors declare no conflict of interest.

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
