# Peer review of "Correlation Analysis of Evapotranspiration, Emissivity Contrast and Water Deficit Indices: A Case Study in Four Eddy Covariance Sites in Italy with Different Environmental Habitats"

_land, doi:10.3390/land11111903_

Round 1
Reviewer 1 Report
1.“remote sensing based methods and approaches have been largely developed and used in order to assess ET over different ecosystems to make up for the lack of ground-based data.” This sentence is vague.
2.“defined as the difference between the surface and dew point temperature” it is unclear description.
3. “for which we observe a linear correlation in between 0.5-0.9. For ECI we have a correlation within 0.5-0.7.“ The correlation range is too large.
4. L310-349 why “Estimate WDI by ET, monitor surface cover by ECI, indicating the correlation between ECI and ET?”
5. L428-429 This paragraph needs to be further detailed.
6. Why does fig1 have no longitude and latitude?
Reviewer 2 Report
land-1927488
The manuscript proposal is good. The authors analysed the relationship between an important variable such as the evapotranspiration (ET) with two indicators, specifically the Emissivity Contrast Index (ECI) and the Water Deficit Index (WDI) for different habitats. For this purpose, in this work data of 4 eddy covariance tower fluxes distributed in Italy, ECMWF meteorological data and satellite images were used. The authors used different statistical scores such as R2 and the Rao’s Q index to study the performance of the selected indicators to represent the bulk of ET and assess habitat heterogeneity. The manuscript is in general well written and the problem correctly stated. However, there are some problems that need to be addressed. I am recommending publication after minor revision.
Specific comments:
Title:
L1: The evapotranspiration process is a sum of soil evaporation and plant transpiration, so I recommend modify the title removing the 'vegetation' word.
Abstract:
It seems that the abstract is more than the required 200 words maximum. Also, the abstract should be a single paragraph.
L49: Please clarify in the text what is the ‘ECMWF’ abbreviation.
L57: Determination correlation or Pearson correlation? Please clarify in the text.
Introduction:
References must be numbered in order of appearance in the tex.
L84: Add ‘s’ to the ‘technique’ word.
L86: [61; 15]
L141 – L157: These lines describe the CAMEL dataset, how it is produced, their main characteristics and how the authors used this product in the current study. I suggest move these lines to the 'Materials and Methods' section.
L184 and 186: Replace ‘evapotranspiration’ for the proposed abbreviation ‘ET’.
Materials and methods:
L200: ‘Middle’ instead of ‘middel’
L255: Monthly ET values are obtained by averaging or accumulating daily values? Please incorporate this information in the text.
L265: In Equation 2 (ECI formulation), the term Ñ” represents the emissivity? On the other hand, between which values range the ECI index?
L285: WDI values are monthly? Same as ECI, between which values range this index?
L294: Why did you select this date (June 2014) to calculate the NDVI?
L308 – Figure 2: Please indicate the date of the selected image.
Results:
L324: What are the principal crops for this study area? Are there crops during the summer months, such as maize? Because Figure 3 show high ET values in summer for the Bosco della Fontana, similar than the other sites. Only bare soil evaporates in these months?
L326 – Figure 3: What is mm/gg? mm/month? The Bosco della Fontana site has no snow cover?
L327: determination correlation?
L331: In Figure 4 the correlation for the Renon site is R2=0.68. Please check this value.
L344: This better correlation could be due to the low number of data points (N=12) compared with sites A, B and C (N>36)?
L348: The analysed indices WCI and ECI can be used as an indicator of the bulk of ET. I suggest change the word ‘estimate’ in the text.
L349 – Figure 5: Sites A, B, and C presents WDI range values different compared with site D (Bosco della Fontana). Instead, Alpine sites presented negative WDI values. What could be said about this aspect? On the other hand, the mean annual air temperature for site D is higher than A, B and C. Could be the air temperature influence in WDI? Please explain these results in more detail.
L349 – Figure 5: In Figure 3 ET observed data is represented by the red line, but in Figure 5 it is in blue. I suggest use the same colour in order to avoid confusion.
L356: Table 1 show that the lower Rao’s Q index is in the Monte Bondone area. Check this sentence.
L362: Monte Bondone is in Figure 7B or 7A? Figure 7 did no match with Figure 2.
L363: right picture??? It seems that both figures are inverted.
Discussion
L372: ET it is an observed data obtained from the eddy covariance towers, so it is not an index. ET is a ground measurement.
L379: Change ‘four’ to ‘4’, or adopted a unique form.
L391: Bare soils evaporate water and can contributes keeping ET in conjunction with the transpiration of the forested area. As was mentioned above, are there summer crops??
L403: Change ‘evapotranspiration’ for ‘ET’ or adopted one of this.
L405: The parenthesis must be closed.
L412: Same as L403
Conclusions
L428 and L433: Same as L403
L442: WDI is an indicator of the bulk of evapotranspiration.
Reviewer 3 Report
Correlation analysis of vegetation evapotranspiration, emissivity contrast, and water deficit indices: a case study in four eddy covariance sites in Italy with different environmental habitats
by
Torresani et al.
General comments:
The study performs evapotranspiration (ET) assessment over four eddy covariance sites with different environmental habitats in Italy. Two indices, emissivity contrast index (ECI) dervied from remote sensing based emissivity data, and water deficit index (WDI) derived from global reanalyses temperature data are presented and compared by evaluating their relationships with ground-based ET measurements. The article is well written, but needs further work. The gap and contribution of the work are not clear. The major methods which the results are based upon, are omitted. The citation arrangement does not follow the standard format of the journal. There are a few typos that should be corrected throughout the text.
In my view, the paper in its current form needs to be updated following the comments, majorly clarifying the purpose of the paper and the methods used to derive the relationship between ET and environmental indices.
Specific comments:
Citations are not well arranged, they should follow in sequence. Please check the journal format and update accordingly through out the text.
Introduction
Lines 88-90: "More recently remote sensing data,..". Remote-sensing based ET studies at different scales span over two decades. Many of these studies including your cited articles are not more recent, please revise the text or supply more recent articles on this topic.
Line 99: "..make use land characteristics as.." should be "..make use of land characteristics such as.."
Materials and Methods
Lines 246-255:. EC measurements are not without limitations, for instance, EC technique generally underestimates turbulent fluxes including ET flux, leading to non-closure of surface energy budget. Other reasons for non-closure have been well-documented (e.g. Foken, 2008). This should at least be highlighted.
Foken, T., 2008. The surface energy balance closure problem: an overview, Ecol. Appl., 18, 1351-1367.
Line 259: ".. to to..". Please correct
Lines 276-281: It is unclear which ECMWF product used by the author?. ECMWF generates products (e.g. ERA-Interim, ERA5, ERA5-Land, etc.) that are fundamentally different. Please be explicit.
This section (Materials and Methods) looks incomplete, for instance, the authors mentioned and presented relationship or correlation between ET and WDI/ECI, however, no method to demonstrate how such results have been achieved.
Additionally, it is unclear how the authors define or characterise WDI and ECI based on ET regimes?
The authors should state the temporal scale in which the assessment are carried out or highlight how they have addressed the temporal mismatch among data products used?
Discussion
Lines 384-385: ...emissivity values. How does the snow cover influence emissivity? this should be discussed.
Lines 417-419: These statements characterising the indices should have also be mentioned earlier in the paper.
Figure
Fig 3-6. Update the axis and legend texts to be clearer.
Round 2
Reviewer 3 Report
Correlation analysis of vegetation evapotranspiration, emissivity contrast, and water deficit indices : a case study in four eddy covariance sites in Italy with different environmental habitats.
by
Torresani et al.
General comment:
The article has broadly been improved and all comments addressed satisfactorily. The article may be published therefore after the minor issue highlighted below has been addressed.
Minor comment:
Lines 248-255, 282-289: Repeated contents. The two paragraphs should be integrated and summarised in just a paragraph.